# A Proposal for Reform of EU Member States' Corporate Governance Codes in Support of Sustainability

**Georgina Tsagas**

College of Business, Arts and Social Sciences, London Law School, Brunel University, Middlesex UB8 3PH, UK; georgina.tsagas@brunel.ac.uk

**Abstract:** An overview of the European Union's varying policies on the harmonisation of Member States' company and securities laws dating back to the 1970s showcases the Commission's averseness to deviate from the path dependence of the shareholder primacy norm and the existence of a series of policies that superficially afford attention to 'stakeholders' rights', 'sustainability' and 'corporate social responsibility'. The article seeks to demonstrate that the 'greenwashing' attempts it identifies in several of the Commission's documents and legislative initiatives have given rise to problematic outcomes, one of which is the subsequent whitewashing of recent initiatives that aim to provide real support to sustainability concerns. The question the article sets out to answer is whether, at this stage in time, the main sociolegal challenges in the form of tensions that the Commission was faced with, in an attempt to address corporate governance at European level in a uniform manner, can now be resolved so as to better support sustainability. If so, what 'softer' options are available to the legislator to signal a renewed approach to corporate governance and a deviation from the path dependence shareholder primacy norm? The argument that the article puts forward is that in order to better complement the latest, more positive attempts, that aim to support corporations' sustainable practices, a reform of European Union Member States' Corporate Governance Codes to include a robust stakeholder friendly provision may well constitute one pragmatic way forward.

**Keywords:** Corporate Governance Codes; EU company law; sustainability; ESG; corporate social responsibility

---

## 1. Introduction

For years, the EU has been supporting studies that aim to compare and contrast the laws and policies between Member States in areas such as the duties of corporate boards, acquisitions, accounting and institutional investor oversight, to name a few. Some have successfully resulted in the adoption of legislative initiatives that further some form of harmonisation in the respective areas, whilst in other areas, little or no progress has been made due to sociolegal, political or practical obstacles that the EU faces from time to time. What is certain, however, is that the future governance agenda on the relationship between corporate governance and sustainability has lacked the required clarity and streamlining necessary. The short-term pressure for maximisation of financial returns to investors, together with a general tendency to see immediate economic growth as a main policy target, have operated as a huge impediment to any serious attempts that would have addressed furthering sustainability objectives more holistically.

An overview of the EU's varying policies on the harmonisation of Member States' company and securities laws dating back to the 1970s showcases the Commission's averseness to deviate from the path dependence of the shareholder primacy norm and the existence of a series of policies that superficially

afford attention to 'stakeholders' rights', 'sustainability' and 'corporate social responsibility'. However, hope can be found in the latest report on Sustainable Finance in 2017, which was released following concerns relating to the assessment of environmental risks in key industries. The Report acknowledges the interplay between corporate governance and sustainability, being key to ensuring that those who lead institutions become fluent in sustainability risks and opportunities. The article seeks to demonstrate that the 'greenwashing' attempts it identifies in several of the Commission's documents and legislative initiatives have given rise to problematic outcomes, one of which is the subsequent whitewashing of recent initiatives that aim to provide real support to sustainability concerns. The question the article sets out to answer is whether, at this stage in time, the main sociolegal challenges in the form of tensions that the Commission was faced with, in an attempt to address corporate governance at EU level in a uniform manner, can now be resolved so as to better support sustainability. If so, what 'softer' options are available to the EU legislator to signal a renewed approach to corporate governance and a deviation from the path dependence shareholder primacy norm? The argument that the article puts forward is that in order to better complement the latest, more positive attempts, that aim to support corporations' sustainable practices, a reform of EU Member States' Corporate Governance Codes with a robust stakeholder friendly provision could constitute one pragmatic way forward.

In an attempt to identify how the shift to the inclusive stakeholders' interests model can be processed, the paper proceeds to identify and evaluate three of the main sociolegal challenges, in the form of tensions that the Commission had been faced with, in an attempt to address corporate governance at EU level in a uniform manner. These are namely the existence of two different socioeconomic and governance models prevalent in the EU, the divergence between company and financial market law initiatives and the limits of harmonisation set by two opposing policy objectives, namely regulatory harmonisation and regulatory competition, respectively. The paper's main contribution is to suggest reform of the Corporate Governance Codes of EU Member States which will, as argued, allow a soft shift in policy towards a corporate governance model that better aligns itself with sustainability, and that truly endorses it as its integral part. An addition of a stakeholder friendly provision that is pragmatic, specific and inclusive is suggested as the way forward.

Existing literature has studied the relationship between corporate social responsibility and Corporate Governance Codes of EU Member States. One study, dated in 2005, provides an empiric study on 22 European Corporate Governance Codes showing that the predominant majority of European Codes orientate themselves to stakeholders and the company. The study stresses, among others, the importance of corporate ethical standards, of dialogue with interested parties and of social and environmental responsibility [1]. Another study, dated in 2013, provides comparative work on the topic and compares the sections that could be identified as stakeholder friendly in EU Member States' Corporate Governance Codes, focusing on ones that stand out for being closer to a stakeholder model of corporate governance. It concludes that, while recognising that there *are* positive exceptions, Corporate Governance Codes are, on a whole, informed by and support the shareholder primacy drive; the Corporate Governance Codes referred to still facilitate and support a system that is based on the externalisation of environmental and social costs of business [2].

The present article relies on the conclusions drawn from the identified comparative work and empirical studies on CSR and Corporate Governance Codes, in order to further the legal discussion on the issue. The topic has recently been addressed from a legal perspective in the work of Sjåfjell who showcases the failure of Corporate Governance Codes to support CSR objectives [3]. The work concludes that attempts at introducing corporate social responsibility language into Codes has been generally superficial, and not designed to achieve the internalisation of externalities that is urgently required if business and finance are to work towards becoming more sustainable, in the economic, environmental and social sense [3]. The present article, hence, takes the discussion a step further, advocating in favour of promoting some form of harmonisation on the issue of corporate governance and CSR at EU level by focusing on the regulatory tool that has been identified as having failed to support sustainable practices, namely the Corporate Governance Codes of EU Member States. This is

done so on the premise that the Corporate Governance Codes can, in fact, signal the desired change of policy towards a stakeholder model and, hence, a proposal for reform which aims to better signal support of sustainability objectives is put forward.

The article's central research question is whether and how the main sociolegal challenges in the form of tensions that the Commission was faced with in the past can now be overcome. Previous aforementioned literature has already engaged with the discussion of CSR and Corporate Governance Codes, but none has proceeded to place particular weight on the background policy tensions that pose an impediment to furthering these objectives, nor has previous literature gone a step further to suggest regulatory reform of the Corporate Governance Codes of EU Member States. To answer the research question effectively, a review of primary and secondary sources on the topics of EU policy on corporate governance, from the 1970s to date, takes place, followed by an identification of key tensions that have formed obstacles to providing a real support to sustainability objectives. With reference to primary and secondary sources, it refers to three distinct initiatives that are recent paradigms of better support of sustainability objectives, namely: Environmental, Social and Governance (ESG) factors, United Nations Development (UNDP) Goals and the EU Non-Financial Reporting Directive. The article identifies key contributions to the debate in the literature and further reviews points raised by the literature focused on comparing the provisions of selected Corporate Governance Codes. The three distinct paradigms chosen by the present article are namely the Danish Code, the Dutch Code and the UK Code, which have been selected on the basis that they represent different perspectives on how sustainability is positioned within the Corporate Governance Codes of best Code for companies.

## 2. The EU's Varying Policies on the Harmonisation of Member States' Company and Securities Laws

Through an examination of the EU's varying policies on the harmonisation of Member States' company and securities laws dating back to the 1970s, the present section seeks to demonstrate that, mainly, due to the Commission's averseness to deviate from the path dependence of the shareholder primacy norm, a series of policies have been brought forward that have superficially afforded attention to stakeholders' rights, sustainability and corporate social responsibility; terms used interchangeably. The identified 'greenwashing' attempts found in several of the Commission's documents and legislative initiatives have given rise to problematic outcomes, one of which is the whitewashing of true sustainability initiatives that have been delivered in more recent years as a response to the climate change emergency. Subsequently, the paper proceeds to identify and evaluate three of the main sociolegal challenges in the form of tensions that the Commission was faced with, in an attempt to address corporate governance at EU level in a uniform manner, and which have arguably prevented it from truly furthering sustainability objectives as well.

### 2.1. Historical Overview Since the 1970s

Ever since the 1970s, the European Community had set out a programme to harmonise the corporate and securities laws of Member States [4]. During the 1970s, the Community focused on the harmonisation of national company laws, following a formal model of harmonisation that was influenced by German corporation law [4]. In the late 1970s, however, the Commission began to focus on the harmonisation of capital markets law instead, following the paradigm of Anglo-Saxon capital market regulatory models. This shift in priorities is an important one for the purpose of the present analysis. Whereas company law attaches importance to the company as a whole, securities markets law attaches importance to investor rights and metrics on financial performance. This shift of focus by the Commission allowed for the tensions that existed between the outsider and insider models of corporate governance between Member States to be resolved by simply giving precedence to the Anglo-Saxon model of corporate governance that supports the shareholder primacy norm. Hence, in the 1980s, the Commission set out to establish equivalent standards of investor protection

throughout the EU, which was conducive to increasing investor confidence in the integrity of financial markets [4]. From the 1990s onwards, regulatory harmonisation and stakeholder protection were treated as an anathema and began to take a back seat in the Commission's corporate governance regulatory initiatives. The promotion of regulatory competition and investor protection was the focus of attention instead [5]. Significant reliance was placed on the ability of an efficient capital market to adjust the terms which governed the relationship of the company towards stakeholders by giving priority to shareholders, with stakeholders' rights being deemed to fall outside the corporate governance ambit [5].

In 1999, the Financial Services Action Plan (FSAP) was introduced, which recognised that the regulation of corporate governance in the EU was becoming an integral part of financial market regulation [6]. The aim of the FSAP was to create a single EU financial market by eliminating market fragmentation and reducing the costs of raising capital [6]. In 2002, the Commission requested from the High Level Group of Company Law Experts, henceforth HLG, to prepare a report on the future priorities for European company law [7]. Winter, the chair of the High Level Group of Company Law Experts, clarified that: "... the primary focus of the EU's involvement in company law should be to establish company law that facilitates efficient and competitive business across the EU, rather than focusing on harmonisation to create similar protection for shareholders and third parties for the sake of it" [8]. As a follow up, in the 2003 Plan on Modernising Company Law and Enhancing Corporate Governance, the European Union adopted an even narrower objective compared to the FSAP, and paid exclusive attention to the protection of shareholders' rights [9]. Proposals included enhancing corporate governance disclosure which would target shareholders' information rights by requiring listed companies to report, in their annual corporate governance statement, on key elements of their corporate governance structure and practices [9].

The emergence of the financial crisis in 2008 prompted a slight shift of focus in policy, which was guided mainly by the need to safeguard market stability [4]. The EU's response to this can be found in the 2011 Report of the Reflection Group on the Future of EU Company Law, which set out to examine what type of deficiencies in EU company law had played a role in the EU's substantial economic downturn [10]. An important aspect of the report concerned the topic of corporate governance and investors' contribution to the long-term viability of companies [10]. Recommendations within this context included reviewing the Corporate Governance Codes and amending them against the background of the rules facilitating a long-term perspective [10]. As a follow up and in light of the social consequences stemming from the 2008 financial crisis, the Green Paper on the EU Corporate Governance Framework [11], dated October 2011, came to approve a new European strategy on corporate social responsibility [12], finding that it was vital to renew its efforts in promoting it. It was stated that the corporation's aim should be one which would maximise the creation of *shared value* for shareholders, as well as other stakeholders and society at large, and to identify, prevent and mitigate their possible adverse impacts [12]. The Report also recognised that in order to further develop its CSR policy, there is a need to adopt a balanced *multi-stakeholder approach*, to clarify what is expected of enterprises, to promote market reward for responsible business conduct, address company transparency on social and environmental issues from the point of view of all stakeholders and acknowledge the role of complementary regulation in creating an environment that prompts corporations to voluntarily assume social responsibility [12]. In December 2012, the Commission produced an alternative plan on a model legal framework for more engaged shareholders and sustainable companies that identified shareholders' short-term investor horizons and shareholders' lack of interest in holding management accountable for their decisions as key problems [13]. The main lines of action all targeted better engaging shareholders in monitoring companies' performance and enhancing transparency by requiring companies to provide better information about their corporate governance to their investors and society at large. No mention was made, however, in relation to the need to clarify the content of that information or its end utility by the constituencies it was assumed to target [13]. Again, the Commission pointed out that the initiatives in the area of corporate governance would not aim at altering the Commission's approach, but rather

at ensuring that the approach becomes more efficient by encouraging proper interaction between companies, their shareholders and other stakeholders.

Against this backdrop of events, the development of EU company law and corporate governance unfolded towards a 'safe approach' and 'familiar direction', despite the mention and reference to key words such as 'stakeholders', 'third-parties', 'long-termism', 'sustainable companies' and 'society at large'. The path followed continued to be that of a minimum standard means of harmonisation and focused on the relationship between the shareholder body and the board of directors. The Commission continued to place trust in the power of financial markets, with the use of metrics deriving from and limited to strict financial performance, as a way of tackling broader corporate governance issues, including short-termism and the protection of stakeholders' interests.

The more recent realisation of the magnitude of the negative impact corporations have on people and the planet has marked a significant shift in the Commission's policy and outlook. In 2017, following concerns relating to the assessment of environmental risks in key industries, the High Level Expert Group on Sustainable Finance issued an interim report on Sustainable Finance, which addressed the topic of investor confidence and corporate governance, acknowledging that the interplay between corporate governance and sustainability is key to ensuring that those who lead institutions become fluent in sustainability risks and opportunities [14]. In this context, it emphasised that investors and lenders need to understand both the risks associated with unsustainable business practices, as well as the interests of their clients in taking account of sustainability considerations and adopting a long-term approach to investment [14]. The experts recommended that: "advisers to institutional investors should have a duty conferred on them to raise ESG issues pro-actively within the advice that they provide and that this requirement would have to be established by the supervisory authorities, based on an extended definition of advisers' duties. More broadly, responsible investment, active ownership and the promotion of sustainable business practices should be a routine part of all investment arrangements, rather than an optional add-on" [14]. On 8 March 2018, the European Commission released an Action Plan for financing Sustainable Growth, in which it examined how to integrate sustainable considerations into its financial policy framework in order to mobilise finance for sustainable growth [15] as a follow up to the recommendations made by the High Level Expert Group (HLEG) on Sustainable Finance. The policy direction following from these initiatives targeted the need to develop and promulgate a set of principles of corporate governance and stewardship that incorporate long-term value creation and improve investor governance. These were seen as central objectives to be furthered at the European level, with sustainability being embedded in the objectives and oversight of the board directors, the investment institutions/funds and their advisers [14,15]. The ways recommended for improving governance included among others: "Developing a set of European corporate governance principles that address long-term value creation and sustainability; and reviewing the potential to incorporate long-term value creation and sustainability as part of the incentive framework in regulated industries" [15].

Key to the Commission's objective of providing a uniform set of rules, which aim to support sustainability in a way that will deliver better results, is to examine how these objectives fit in with the Commission's current action plan. The Commission's Action plan includes ten action plans in total. The tenth one specifically refers to: "fostering sustainable corporate governance and attenuating short-termism in financial markets" [15]. Action 10 specifically provides that the Commission will be carrying out: " . . . analytical and consultative work with relevant stakeholders to assess: (i) the possible need to require corporate boards to develop and disclose a sustainability strategy, including appropriate due diligence through the supply chain, and measurable sustainability targets; and (ii) the possible need to clarify the rules according to which directors are expected to act in the company's long-term interest" [15]. Action 10 in combination with Action 1 on establishing an EU classification system of sustainable activities showcases the EU's intention of reinforcing policy towards the direction of integrating sustainability as a core aspect of corporate governance. The latest policy objectives brought forward by the Sustainable Finance initiative show a more ambitious strategy on the Commission's

part as they provide for concrete steps to align sustainability objectives with corporate governance ones. There is now clearly a shift in priorities on what constitutes 'good practice' from a corporate governance perspective. Good governance is one which does not only endorse sustainability but one which also includes it.

The present overview shows that the EU's seemingly changing outlooks on issues relating to corporate governance, stakeholders and sustainability were, in reality, a reproduction of the same approach; that of the shareholder primacy norm. The window-dressing of policy documents with key wording on related to CSR and sustainability issues may have given the illusion of a change in relation to the Commission's outlook. On a positive note, however, in recent times, shifts and changes in the Commission's policy documents can be identified, which undoubtably showcase a disposition to the adoption of a policy that not only endorses sustainability but includes it as an integral part of a corporation's good governance. In order to better estimate future options available, the following section proceeds to identify and evaluate three of the main sociolegal challenges in the form of tensions that the Commission had arguably been faced with when regulating corporate governance at EU level in a uniform manner. These tensions had, as the paper argues, prevented it, to a certain extent, from truly furthering sustainability objectives in the past. An examination on how these tensions can be addressed so as to move forward will be further discussed.

*2.2. Tensions Regulating Sustainability Issues at EU Level*

As early as 2005, Commissioner McCreevy had stressed the importance of legal certainty for the proper operation of markets by stating that: "Without legal certainty, without reliable information, without clear framework rules, markets cannot work for long" [16]. Tensions, as between different models prevalent in the EU, as well as different regulatory objectives have compromised the EU regulator's aims of achieving consensus on a uniform set of rules in particular fields of EU law. Corporate governance has proven to be an especially problematic case in this respect. The questioning of the Anglo-Saxon corporate governance model as a consequence of the emergence of the 2008 financial crisis, and the more recent common realisation of the magnitude of the negative impact of corporations' business practices on people and the planet, have arguably prompted a change in priorities. As mentioned above, the Commission Sustainable Finance Action Plan 10 specifically aims to look into options that would enable corporate boards to develop and disclose a sustainability strategy, including appropriate due diligence through the supply chain, and measurable sustainability targets; and options that would help clarify the rules according to which directors are expected to act in the company's long-term interest. In view of the Commission's 2018 renewed commitment to sustainability, it is worth exploring whether the Commission can, in fact, henceforth achieve the necessary political support to further its objective of providing a harmonised regulatory framework on supporting sustainable practices through uniform provisions within the corporate governance framework. The present section hence identifies and evaluates three of the main sociolegal challenges in the form of tensions that the Commission was faced with in an attempt to address corporate governance at EU level in a uniform manner and which, as it is argued, have prevented it from truly furthering sustainability objectives. It also first proceeds to identify issues relating to the interpretation of the term 'sustainability' and associated terms, such as CSR and stakeholders' interests.

2.2.1. Definitions and Terms: Sustainability, CSR and Stakeholders' Interests

A tension exists between different understandings of sustainability. The terms 'sustainability' and 'sustainable development' from a policy perspective, 'sustainable development' from a business perspective, 'environmental social governance (ESG)', 'greening', 'triple bottom' line [17] and 'corporate social responsibility (CSR)' are often used interchangeably. Management literature has, since the 1990s, made attempts to incorporate notions of sustainable development into corporate strategy and has discussed the emergence of corporate environmentalism and organisational processes of environmental management [18]. The S&P Dow Jones Index, which is managed cooperatively by S&P

Dow Jones Indices and RobecoSAM, defines 'corporate sustainability' as "a business approach that creates long-term shareholder value by embracing opportunities and managing risks deriving from economic, environmental and social developments" [19]. The ICC Business Charter for Sustainable Development 2015 draws a distinction between 'Sustainable Development' from a policymaker's perspective by referring to the definition provided by the 1987 Brundtland Report, and 'sustainability' or 'sustainable development' in a business context, by referring to it as a process whereby companies seek to manage their financial, societal (including governance) and environmental risks, obligations and opportunities, otherwise known as a 'triple bottom line' approach [20]. The ICC Business Charter also points out that the term 'sustainability/sustainable development' is often used as an umbrella term including 'Corporate Social Responsibility (CSR)', 'Environmental, Social, Governance (ESG)' or 'triple bottom' line [20]. Acknowledging this distinction, the European Commission makes reference to 'sustainability' within the field of corporate governance in its latest reports as synonymous to a company's long-term business growth [21]. As a term, 'sustainability', has come to be a contested concept. In regulating for sustainability, which for the purposes of our discussion in the present article includes sustainability in the economic, environmental and social sense, a major challenge arises due to the multiple and competing definitions of the terms 'sustainability' [22] and that of 'climate change' [23], which entail geographical distinctions and diverse understandings of sustainable development.

In 1987, the World Commission on Environment and Development, in what is widely known as the Brundtland Report, defined development as 'sustainable' when it "meets the needs of the present without compromising the ability of future generations to meet their own needs" and described it as "a process of change in which the exploitation of resources, the direction of investments, the orientation of technological development; and institutional change are all in harmony and enhance both current and future potential to meet human needs and aspirations" [24]. The Brundtland Report's definition of 'sustainable development' underlines that the relevant 'needs', are "the essential needs of the world's poor, to which overriding priority should be given; and the idea of limitations imposed by the state of technology and social organisation on the environment's ability to meet present and future needs" [24]. Gray suggests that the Brundtland Report's definition has led to "a widespread agreement" that sustainable development "involves the preservation and/or maintenance of a finite and crucial environment; and incurs some duty of social justice—between and within generations" [25]. In this context, the scientific contribution of the notion of 'planetary boundaries' becomes useful, without which, regulating would have no meaningful target [26,27]. The scientific work introduced a new approach to global sustainability in which planetary boundaries are identified and defined as ones in which they expect that humanity can operate safely, done so with an aim of laying the groundwork for shifting the approach to governance and management, away from the essentially sectoral analyses of limits to growth aimed at minimising negative externalities, toward the estimation of the safe space for human development [26,27].

The use of the term 'sustainable development' has also been used much in international initiatives. The term was adopted as an overarching objective by Governments at the Earth Summit of 1992 in Rio de Janeiro, together with a set of Rio Principles and a global action plan, Agenda 21 [28]. The United Nations has also promoted the objective of sustainable development, recognising throughout its reports that strong interdependencies exist among the economic, social and environmental dimensions of sustainable development [29,30]. In 2015, Member States of the United Nations adopted a set of Sustainable Development Goals to end poverty, protect the planet, and ensure prosperity for all, as part of a new UN sustainable development agenda, encouraging governments, the private sector, civil society and individuals to participate in the realisation of these goals [31].

As mentioned, at the EU level, the European Commission's reference to sustainability within the field of corporate governance can be observed in its 2011 report as synonymous to a company's long-term business growth [11] and the Treaty of Lisbon also refers to the term 'sustainable development', but in an EU context. Article 3, paragraph 3 of the Treaty of Lisbon, states, among others, that: "The Union shall establish an internal market. It shall work for the sustainable development of Europe based on

balanced economic growth and price stability, a highly competitive social market economy, aiming at full employment and social progress, and a high level of protection and improvement of the quality of the environment." The Article's structure, as it has been argued, implies that the promotion of the social objectives outlined in paragraph 3 is seen as a goal of the Union on a par with the establishment of the internal market, rather than being viewed as a consequence of it, as the old article 2 TEC implied [32]. Bruun et al. argue that the provision should thus be read as encouraging the taking of active steps to promote social goals separately from those taken to establish the internal market [32].

A distinct divergence exists between terms in relation to whether sustainability is understood as planetary or community sustainability or rather the sustainability of the corporation through its long-term growth. This qualifies as the key difference considering that different understandings of sustainability within this context give rise to different priorities: addressing risks within the context of the planetary boundaries [33], including environmental, human and social risks, or risks to the microcosmos of the corporation or an industry. The present article refers to the term, as elaborated on by Gray, namely that sustainability is connected to ecological and societal boundaries that are not necessarily the same as organisational or corporate boundaries [25]. Gray explains that the problem lies in the fact that an understanding of sustainability is largely a collective outcome of personal value judgements around politics, nature, religion, planetary ecology and morality; and sustainability may rely on overall interactions within a broader system that cannot easily be predicted [25]. Despite this reality, regulation can come to carefully address aspects of sustainability and proceed to close rather than open the wide gap that exists in relation to its vague understanding through the use of definitions in the law. What also needs to be taken into consideration is sustainability in the broader sense, inclusive of planetary boundaries, rather than the microcosmos of the corporation or an industry. In this respect, the EU's definitions, as outlined in key documents and legislative initiatives and policies, need to be revisited.

### 2.2.2. Different Types of Socio-Economic Models

The tension which exists between the two different systems of corporate governance and economic systems prevalent in the EU has been an obstacle to reaching a consensus on regulation on sustainability within the context of corporate governance. The two different systems are namely the insider and the outsider system of corporate governance. Germany and France, for example, are countries with insider systems, whereby shares are concentrated in the hands of a small number of investors and blockholders, who build strong relationships with the management of the company and have an incentive to monitor management due to the large stock they own [34]. Traditional outsider systems in the EU are the UK and Ireland. Companies in these systems have dispersed ownership structures and relationships between shareholders, and managers are loose. These systems usually have an active market for corporate control, in which takeovers will work against management inefficiency and align managers' and shareholders' interests [34]. The Anglo-Saxon model gives priority to shareholders' welfare based on the notion of ownership rights, whilst the Continental European one, aims to balance out broader interests within the company and affords special attention to employee rights and their participating in the company's decision-making process [35]. The Continental European idea of corporate social responsibility and the respect afforded to employees is not at terms with the idea of strict shareholder wealth maximisation prevalent in Anglo-Saxon model.

The two different systems of governance have developed as part of two different broader types of socio-economic systems dominant in the EU. The 'varieties of capitalism' approach considers companies as the crucial actors in any capitalist economy, and assumes, as the term suggests, that within the EU, a variety of capitalist economies with different characteristics coexist. The 'varieties of capitalism approach' draws a distinction between liberal market economies, henceforth LME, and coordinated market economies, henceforth CME [36]. In the former, "firms coordinate their activities primarily via hierarchies and competitive market arrangements" [36], whereas in the latter "firms depend more heavily on non-market relationships to coordinate their endeavours with other actors and to construct

their core competencies" [36]. The existent monitoring systems within each economy also determine the company's access to finance [36]. In LME, investors rely on the information provided in balance sheets so as to monitor the progress of the company they have invested in. Markets are the institutions that firms will primarily rely on [36]. In LME, firms focus on their share price in the stock market and are subject to hostile takeovers, which may take place when the share price of a company is low [36]. Conversely, in CME, the financial system provides companies with access to finance that is not dependent on publicly available financial data, but on private or otherwise termed 'inside' information about the company [36].

The continuous battle between the Anglo-Saxon and Continental European models had been one of the most important variables affecting the Community's ability to adopt a uniform legal framework on aspects of company and financial markets law and, subsequently, on sustainability within the corporate governance context.

### 2.2.3. Company and Capital Market Law Objectives

According to Heiser, the aim of company law is threefold: to create a general constitution for corporate behaviour, to take account of interests of third parties holding a stake in companies' activities, and to ensure the economic feasibility of acting in the form of a corporation [37]. Company law rules therefore address, among others, the means via which to achieve a corporation's long-term growth and the protection that needs to be afforded to third parties' interests. Capital market law, on the other hand, is to be understood as: " . . . the totality of all legal rules and principles that regulate the flow of financial means between the investor, as supplier of his capital, and the undertakings, as its demanders and suppliers of companies, government, or private securities in return" [37]. The tension that exists between capital market law and company law is furthermore elaborated on by Heiser, who explains that capital market provides protection to investors so as to secure the functionality of the market, whereas company law focuses on the long-term realisation of specified economic goals by searching for a balance between different kinds of conflicting interests [37]. It is obvious that there is a divergence between the relationships and interests which each law aims to regulate and protect, respectively. Capital markets law aims to facilitate well-functioning capital markets and adds emphasis to investor protection, whereas company law covers matters related to corporations' establishment, organisational structure, as well as corporate constituencies' rights and duties and often regulates the links that exist between them [5]. Whereas securities law places investor protection as its primary objective, company law protects, or at least recognises, the protection of the company as an entity and affords particular attention to the notion of the corporate interest acknowledging the contribution of the inputs of other constituencies in achieving the goal of the company's long-term continuity.

Corporate Governance Codes arguably fall within the ambit of the two areas of laws, as they target *company law* and *financial law* issues of listed companies. The difficulty in reaching a balance between meeting the objectives of these two areas of law can be narrowed down to the simple fact that capital markets law regulates listed companies and aims to foster the free transferability of shares within the market place, whereas company law acknowledges the need to protect corporate policy and to foster the corporations' long-term growth. Company law and capital markets law have a different set of legal objectives, which may also vary from one EU Member State to another. The fact that there is no clear standpoint as to the corporate governance model that the Commission supports in either areas of law, whether company and/or capital market law, has arguably created problems in adopting a uniform set of rules on sustainability issues as well.

### 2.2.4. Harmonisation and Its Limits

Another important question that needed to be considered is how much harmonisation is in fact necessary for the effective promotion of sustainable practices. In order to answer the question, consideration needs to be given to the topic of what the right balance that the Commission needs to strike between creating a level playing field through uniform standards for companies on the one hand

and ensuring that national diversity is respected on the other. The tension is basically reflected in the fact that the interests of the Member States are to protect local markets through sector self-regulation, as compared to those of the community, which are to protect the market at large through regulatory harmonisation [38,39].

The proper function of capital markets is an important aspect of financial integration within the EU. The 2003 Modernisation of Company Law Programme made clear that the Commission aimed to adopt rules that would strike a proper balance between actions at EU level and actions at national level [40]. A certain level of regulatory competition and a certain level of regulatory harmonisation was assumed to strike that balance. What was however left unclear was how one would determine *which* rules were likely to be best dealt with at *EU level* and which more efficiently at *national level*. The tension between regulatory harmonisation and regulatory competition is exacerbated in the EU by the fact that there is a grey area as to what needs to be regulated at Member State level and what at EU level. Tensions as such, coupled with the difficulty in reaching a political consensus on a mutual approach towards a particular set of laws, had led the EU to adopt a 'minimal standards' approach to the regulation of many aspects of the regulation of corporate governance and financial markets law. The tension between regulatory harmonisation and regulatory competition initiatives [41,42] and the test stemming from the EU principle of subsidiarity [43] have sought to address these issues. Economists and supporters of the law and economics movement are, in general, against harmonisation [44], whereas European law specialists are in favour of it [45]. Regulatory competition is supported on the grounds that it allows Member States to compete with one another in the kinds of laws offered in order to attract businesses or investors and depends upon the ability of such actors to move between two or more separate legal systems. Regulatory competition may create a 'race to the top' or a 'race to the bottom' in standards. There are, however, good arguments in favour of harmonisation. The application of maximum harmonisation standards is beneficial insofar as it increases uniformity and legal certainty [4]. Combating regulatory differences through the adoption of uniform securities laws also has the benefit of reducing transaction costs for issuers of securities who want to offer their securities across countries, as well as for investors who are able to compare investment opportunities in different countries [46]. Uniform rules also allow for the creation of a level playing field, as they guarantee equal opportunities to issuers and investors, placing economic performance as the sole measure by which their performance is assessed [46].

Regulatory harmonisation, however, can also result in negative side effects for Member States and the business community, as it may unjustifiably disregard important national differences and hamper the advantages of regulatory competition, i.e., more choices to issuers and investors, as well as allowing for the best law to prevail amongst regulators through a 'race to the top' [46]. However, regulatory competition may have the adverse effect and lead regulators to a 'race to the bottom' rather than to a 'race to the top', as investors may fail to identify the legal system which is the best or may opt for and encourage a system which simply offers the most lax rules [46]. This scenario is even more true in relation to the protection of stakeholders' interests within the context of sustainability.

Regulatory competition, in the form which has opened within countries that have federal systems of regulation, such as the United States of America and the Delaware-style scenario that applies within, is highly unlikely to apply in the EU, however, as reincorporation is difficult and costly [47]. In the EU, extreme regulatory competition is hard to exist since, as McCahery and Vermeulen explain, the company law measures already in place prevent Member States from exercising a high degree of autonomy about corporate governance regulation overall [48]. The trend in the EU has so far been towards uniformity rather than maximum harmonisation [4]. Uniformity is assumed to be achieved through minimum harmonisation standards set at EU level allowing Member States to provide more stringent rules [49]. However, uniformity through the means of minimum harmonisation is not considered the ideal approach for areas that form the core of creating the European single market. The harmonisation of securities and company laws, including Corporate Governance Codes, qualify as such areas, making it necessary for maximum and not minimum harmonisation standards to apply.

The principle of subsidiarity is addressed in Article 5 of the EC Treaty [50] and aims to ensure that decisions are made as closely as possible to the needs of citizens and that checks are made in order to establish whether action at Community level is justified [51]. According to the principle of subsidiarity, the EU does not take action, with the exception of the areas that fall within its exclusive competence, unless it is more effective than action taken at the national level [51,52]. Conditions of compliance with the subsidiarity principle are obscure [52]. The ambiguity surrounding the interpretation of the subsidiarity principle makes it difficult to establish whether or not proposed legislation complies with the principle of subsidiarity. This could work positively in favour of regulating sustainability at EU level. As Papadopoulos contends, with reference to the subsidiarity principle, there is only a need for the EU legislator to better justify the need for community action, especially for detailed and intrusive provisions, and in doing so, reflect on measures which promote and secure the coherence of the internal market, which can only be achieved at Community level [51]. On the topic of sustainability, a common ground *can* be found between Member States, supported by the recent global collective call for action on the planet's climate emergency and following the Sustainable Finance initiatives.

### 2.3. EU policy on Sustainability: Present

The overview of the EU's policy documents aimed to provide a picture of the approach that the EU has taken towards EU company and financial markets law in recent times [53]. With the introduction of sustainable finance initiatives, there has been a marked shift in the Commission's approach towards the regulation of company and capital markets law in general, which will affect, henceforth, the way in which the regulation of sustainability within the corporate governance context will be addressed in the years that follow. The shift in policy objectives leads the article to support that the approach taken in Corporate Governance Codes on issues of sustainability is anachronistic, and that the adoption of a uniform set of rules to set the tone towards the adoption of a corporate governance model that is inclusive of stakeholders' interests, CSR and sustainability is necessary.

The Commission's position, despite theoretically aspiring to support stakeholders' interests in the corporate governance framework at an EU level, has only in fact managed to adopt a minimum harmonisation approach towards the regulation of corporate governance by affording exclusive attention to the protection of shareholders through the main regulatory initiatives, to the exclusion of the protection of the corporate interest and other constituencies. The aim of the Commission was not to create "a level playing field among EU countries, so that market success is the judge of different varieties of capitalism", but rather to force all European economies in adopting the Anglo-Saxon model of corporate governance [54]. The harmonisation of capital market and company laws should aim to initiate convergence to the best standard, which if adopted uniformly all throughout the EU, would, in theory, enhance the EU's financial development [54]. However, convergence has not come in the form of harmonising the divergent rules of Member States on the subject matters addressed, but rather in the form of forcing particular Member States to converge to the rules of Member States which had provided for a liberal-open market outsider model of corporate governance. The Commission had been criticized for adopting such an approach towards the regulation of corporate governance at EU level. The European Trade Union Confederation, for example, had identified early on that there was a problem in the Commission's policies, insofar as governance was narrowly focused on the relationship between shareholders and management, neglecting the interests of other constituencies of the company [55]. A 2006 Resolution by the European Trade Union Confederation emphasised the need for the European corporate governance framework to: " . . . lay down proper institutional conditions for companies to promote long-term profitability and employment prospects, define mechanisms that prevent mismanagement and guarantee transparency and accountability with regard to investments and their returns" [56]. The European Parliament had also called on the Commission to take the European social model into consideration when deciding on further measures for the development of company law, which also involved employee participation [57]. The narrow focus on shareholder rights and investor protection was arguably not reflective of the divergent socioeconomic models or

the varieties of capitalism prevalent in the EU. It was also not reflective of the role that company law objectives should play in the drafting of corporate governance laws at EU level.

The financial crisis of autumn 2008 followed by the more recent realisation of the magnitude of the negative impact corporations have on people and the planet, have come to challenge the notion that the laws' sole objective is that of regulating the principal-agent relationship and have come to challenge the orthodoxy of the 'minimal guidelines' approach followed by EU legislators. Concerns relating to the assessment of environmental risks in key industries have built awareness of the problematic policies in place even more. They have pointed out that the survival of corporations has an element of public interest in it, as the collapse of certain corporations has unexpectedly and significantly affected national economies, the welfare of stakeholders, society as a whole, and people and the planet overall. The Commission's failure to regulate corporate governance at EU level effectively is a sign that there is a need to adopt a more flexible and efficient approach towards the regulation of corporate governance, by encouraging proper interaction between companies, their shareholders and other stakeholders. Hence, the regulation of corporate governance at EU level should be guided by higher levels of harmonisation of corporate governance standards and consideration should be given to the adoption of rules which respect companies' long-term sustainable growth and the interests of other corporate constituencies in a holistic way. The provisions of most Corporate Governance Codes of EU Member States are arguably not reflective of how business should be regulated in the 21st century, whereby a more enlightened shareholder value approach should prevail. Despite the fact that the Commission has *not radically* changed its overall approach, it does acknowledge that there *is* a need for the previous approach it had adopted to become more efficient by encouraging interaction between companies, their shareholders and other stakeholders.

## 3. Corporate Governance Codes, Disclosure and Sustainability

The United Kingdom was the country which was the pioneer in introducing guidelines on what constituted good practice for companies by launching the UK's Cadbury Report produced by Sir Adrian Cadbury in 1992 [58]. This first step on the road to the initial iteration of the Code was a response to major corporate scandals associated with governance failures in the UK. Following the UK's paradigm and since the 1990s, Corporate Governance Codes started being developed in various countries and were issued from stock exchanges, corporations, institutional investors, or associations (institutes) of directors with the support of governments and international organisations [59]. Key to this development worldwide was also the proposal of *The Organisation of Economic Co-operation and Development (OECD)* in 1998 to develop global guidelines on corporate governance and encourage States to introduce such guidelines [60]. The OECD Principles have not remained static. Specifically in relation to sustainability, recent updates in 2015 have come to "embrace the shared understanding that a high level of transparency, accountability, board oversight, and respect for the rights of shareholders and role of key stakeholders is part of the foundation of a well-functioning corporate governance system" [61]. In 2015, the OECD published the G20/OECD Principles of Corporate Governance. These contain the results of the second review of the Principles which had been conducted in 2014/15 and which place, among others, focus on the role of stakeholders in corporate governance with emphasis on the need for the framework to recognise the rights of stakeholders, which are already established by law or through mutual agreements and encourage active co-operation between corporations and stakeholders in creating wealth, jobs, and the sustainability of financially sound enterprises [61]. Corporate Governance Codes worldwide, most of which followed the UK paradigm, have been focused at their core on regulating the relationship between managers and shareholders with the objective of safeguarding shareholders' rights. However, examples did exist in which the interests of stakeholders, as well as shareholders, were recognised, namely in King I, the Corporate Governance Code of South Africa, in 1994, which was the first Code to embrace the rights of both constituencies [62].

Corporate Governance and CSR, as Szabo and Sorensen point out, were originally considered to be distinct in terms of their objectives. More specifically, corporate governance was originally intended

to address the internal dynamics of the company, limited to the relationship between the shareholder and director bodies, whereas CSR, partly oriented towards employees, mainly focused on the external company's affairs and how the company's operation affected wider society [2]. The shareholder primacy model endorsed for decades had gradually started being superseded by the stakeholder one, so that CSR now is seen to comprise of both an internal and an external element [2]. Until 2013, the EU Commission had not taken any major steps to align CSR objectives with corporate governance ones, but had proclaimed at that time that a launch of an ambitious strategy along those lines was on its way [2]. The overview in Section 3 above, showcases, however, that prior to the Sustainable Finance Report, a larger strategy that would set in place specific legislative initiatives targeting sustainability had not, in fact, been set in place. So, despite the 2011 Commission's revised definition of CSR proclaiming that a corporation's aim should be that which would maximise the creation of shared value for shareholders, as well as other stakeholders and society at large, and to identify, prevent and mitigate their possible adverse impacts, concrete steps towards removing the shareholder primacy norm from key legislative initiatives has not been undertaken.

Hence, with the shareholder primacy norm still steering the Commission's legislative initiatives, it continues to be the case that Member States that do adopt a shareholder primacy approach to corporate governance are likely not to have Corporate Governance Codes that truly endorse sustainability as an efficient form of best practice. This is because, according to this school of thought, it does not constitute part of corporate governance at all and falls 'outside' the ambit of what concerns the corporation. EU policymakers proclaim, indeed, that governance that is narrowly focused on the relationship between shareholders and management is no longer one which is to be considered good practice. However, there is no explicit endorsement of the insider model of corporate governance, nor of the European social model when deciding on further measures for the development of company law. The response followed by the EU until recently, in light of the realisation that unsustainable practices among corporations need to be addressed, is one which is typical of the Anglo-Saxon models of corporate governance and relates to addressing the problems by imposing a requirement of corporations' disclosure of information relating to their impact on stakeholders.

Disclosure, hence, became the popular and conventional regulatory strategy in the business arena to help assess the performance of corporations in the marketplace and largely operates as a market discipline tool, by targeting the investors as the priority recipient group. The relevant information is seen to be useful to investors, since based on the information, they can make profitable investment decisions that will help efficient companies to thrive in the market and punish companies that are not well managed. Financial reporting has been a long-established tool for accountability and a similar approach has also been increasingly adopted for non-financial impacts of corporate activity [63]. However, in supporting efforts to reform the law towards sustainability objectives, a requirement set for corporations to produce reports on sustainable practices or impact on stakeholders constitutes a meaningless task, as it lacks the required infrastructure to achieve real results. This is because, as Ha-Joon Chang has rightly identified in relation to the problems pertaining to capitalism and CSR, including the workings of the market as such: "People 'over-produce' pollution because they are not paying for the costs of dealing with it" [64]. The problem lies in the fact that market pricing is unlikely to reflect the sustainability of a corporation within its broadest sense, since all relevant information on CSR practices is not reflected through systematic reporting, due to the fact that companies do not yet systematically report on their People and Planet impacts [65,66]. Hence, the non-financial reporting requirement may, at instances, be seen as demanding performance of an activity that is manifestly pointless because of a lack of comparability, accountability and impact; it may leave corporations meaningfully disengaged from the overall process, even if they are otherwise in favour of opting for being more sustainable in their operations per se.

Both financial and non-financial reporting at EU level essentially target shareholders with the purpose of enabling them to make informed investment decisions; sustainability or corporate social responsibility (CSR) reporting, however, targets multiple stakeholder groups and informs the wider

community. A key part of CSR reporting, which includes companies' objective of displaying its responsibility towards a wide range of stakeholders, responds to stakeholders' expectations. With CSR reporting, a company manages its own legitimacy, guarding its reputation and identity by engaging with stakeholders. The use of two specific terms make it clear that the shareholder primacy norm has been the one which steers the legislative initiatives at EU level. The concept of 'materiality' and the concept of 'risk management' both essentially target shareholders' interests.

The concept of 'materiality', referred to in the Report on Materiality by the Integrated Reporting Committee states that: "The interpretation of materiality varies across report forms due to differences in audience, purpose and scope. In Integrated Reporting, a matter is material if it could substantively affect the organization's ability to create value in the short, medium or long-term. The process of determining materiality is entity specific and based on industry and other factors, as well as multi-stakeholder perspectives" [67]. The Guidelines Communication on the NFRD relating to the disclosure of material information, reflecting on Article 1 of the Directive, provide that materiality of information must be assessed in context and that materiality will be company specific, which will be assessed by the company itself. Reference to 'materiality', despite being useful insofar as it respects the characteristics of each company in terms of its sustainability operations, also leads to the negative outcome of not providing metrics on sustainable practice as an integral part of good governance more broadly for people and the planet.

'Risk management' is a key term which, in certain respects, has been thought of as a way to provide a means of bridging the gap between external and internal governance of the company. However, the finance literature describes risk management as being concerned with identifying and managing a firm's exposure to financial risk [68]. Kaen, however, in his research, explains that risk management needs go beyond this narrow understanding provided by finance scholars [69]. As Kaen explains: "Corporate governance is often described as the set of rules, structures and procedures by which investors assure themselves of getting a return on their investment and ensure that managers do not misuse the investor's funds ... " and his work " ... addresses the connection between risk management and corporate governance and the public corporation" and argues that "risk management and risk management products help ensure the survival of the firm and thereby support broad public policy objectives—objectives beyond the immediate interests of the owners of the company and a narrow financial objective of shareholder wealth maximization" [69]. Confusion, at present, exists; as in certain instances, what is labelled as CSR or sustainability, is in fact equated to a way of managing risk for the organisation and ultimately for shareholders' interests, following the interpretation given by finance scholars. Hence, risk management has to either be seen in a new light or as a notion which is separate from pre-existing definitions which link risk only to shareholders' interests and not to those of stakeholders' interests, and people and planet.

## 4. Corporate Governance Reform Proposals

### 4.1. Complementary Initiatives: International and EU Initiatives

The Principles for Responsible Investment, the United Nations Development Programme Finance Initiative and the Generation Foundation produced the 'Fiduciary Duty in the 21st Century Report' published in September 2015 with the purpose of providing conclusive comments on the debate about whether fiduciary duty is a legitimate barrier to investors to integrating environmental social and governance (ESG) issues into their investment processes [70]. However, at an international level, progress on research relating to Environmental, Social and Governance, and the duties of financial intermediaries within this context more specifically, had been made as early as 2005 by the United National Environmental Programme (UNEP) jointly with Freshfields Burckhaus Deringer LLP [71]. The 2005 report commissioned by UNEP FI, from law firm Freshfields Bruckhaus Deringer, concluded that integrating ESG considerations into investment analysis is "clearly permissible and is arguably required" [71].The Fiduciary Duty Country Roadmaps were based on interviews with

investors, industry associations, lawyers and policymakers in eight countries and examined the relationship between sustainability and the prudent management of capital. The report concluded that failing to integrate long-term investment value drivers, including ESG issues, in investment practice, constitutes a failure of fiduciary duty. The 2015 Report advocated that action is needed to modernise definitions and interpretations of fiduciary duty in a way that ensures these duties are relevant to 21st century investors [70]. The report proceeded to propose a series of recommendations for policymakers and regulators to consider, which included for them to: " ... support efforts to harmonise legislation and policy instruments on responsible investment globally, with an international statement or agreement on the duties that fiduciaries owe to their beneficiaries. This statement should reinforce the core duties of loyalty and prudence, and should stress that investors must pay attention to long-term investment value drivers, including ESG issues, in their investment processes, in their active ownership activities, and in their public policy engagement" [70].

The growing consensus on the importance of Environmental, Social and Governance (ESG) among investors, is one which was pointed out by Larry Fink in his letter in January 2018 "Letter to CEOs" stating that: "To sustain that [financial] performance ... you must also understand the societal impact of your business as well as the ways that broad, structural trends—from slow wage growth to rising automation to climate change—affect your potential growth" [72]. Ratings of companies on ESG are provided by a range of agencies, such as FTSE4Good, MSCI ESG, Sustainalytics and ISS. However, ratings from such agencies have been criticised for suffering from diverging standards and 'inherit biases', reference being made to the fact that there is a tendency to award higher ESG grades to the biggest companies or ones based in countries with hefty reporting standards [73]. The randomness attached to the collecting of data on ESG available, rather than data on the true ESG impact of a company is a huge deficiency which various flawed regulatory initiatives exacerbate. Corporate Governance Codes lacking consistency among each other on a harmonising approach to sustainability, as well as the reality of policy incoherence on the topic of sustainability, reinforce the problem of a lack of reliability and comparability of sustainable practices among companies.

The 2015 UNEP Report provides a forward-looking approach, as it set out to clarify investors' obligations and duties in relation to the integration of environmental, social and governance (ESG) issues in investment practice and decision-making [70]. The Report identified a class of investments that could reasonably be assumed offensive to the average beneficiary, for example, where there are clear breaches of widely recognised norms, such as international conventions on human rights, labour conditions, tackling corruption, and environmental protection; such that they could lawfully be excluded from an investment portfolio without all the beneficiaries' express consent [70]. Thus, the report is useful, insofar as it has identified a common ground in terms of shared values recognised globally that could feature as a starting point in legislative initiatives that aim to target companies in different industries and sectors and is not limited to company specific ESG factors.

Moreover, in relation to non-financial reporting in the EU specifically, the EU Non-Financial Reporting Directive requires, since 2018, around 6000 large firms to include *non-financial statements in their annual reports* and to disclose relevant information on environmental and social aspects, including reporting on due diligence processes as well as the risks of environmental and social impacts with regard to an entity's own operations or products, services and business relationships. The Guidelines provided by the Commission on non-financial reporting, more specifically on the methodology for reporting non-financial information published in 2017, has proven to be more elaborate on issues relating to implementation and considering of various standards [74]. Additionally, in 2015, after three years of consultations and negotiations, all 193 UN Member States of the United Nations adopted the 2030 Agenda for Sustainable Development, which contains 17 Sustainable Development Goals (SDGs) and 169 targets [75]. As provided by the UN: "the Sustainable Development Goals are the blueprint to achieve a better and more sustainable future for all. They address the global challenges we face, including those related to poverty, inequality, climate change, environmental degradation, peace and justice" [75]. The international initiative on SDGs is considered very important, because,

as stated, it helps with the fact that it acknowledges that: " . . . the relevant (or 'material') impacts in the context of the SDGs are not limited to topics that have a significant financial impact on companies. Rather, the SDGs require a new materiality threshold that includes a wider set of economic, environmental and social impacts and is set from the perspective of SDG stakeholders, not a company's bottom line" [76]. Furthermore, SDGs help appreciate that there exists alternative ways of doing business which can gradually be introduced in companies that are moving their business models in a more sustainable direction [76]. The outline of these recent initiatives shows that there exist initiatives that support sustainability at EU and at an international level and, hence, the proposal for reform introduced by the present article aims to align itself better with the policy the EU needs to adopt on sustainability more generally.

### 4.2. Corporate Governance Codes in EU Member States: Paradigms

Wieland, in 2009, identified Romania, Belgium, France, Germany, Italy, Russia, Austria, Denmark, Hungary, Poland, Slovakia, Lithuania, Spain and Turkey as countries that adopt a stakeholder orientation; and Switzerland, Czech Republic, Portugal, Sweden, Finland, Great Britain and Ireland as countries with a shareholder orientation [1]. A useful comparison of EU Corporate Governance Codes in relation to their approach to CSR has also been provided by Szabó and Sørensen in 2013 [2]. Szabó and Sørensen single out, among various Corporate Governance Codes in Europe, the Danish Code, the Icelandic Code and the Greek Code, as ones which adopt a clear stakeholder policy recommendation towards company boards within their text [2]. The work of Szabó and Sørensen, who investigate and analyse how CSR issues have been integrated in Corporate Governance Codes [2], provides important conclusions regarding the effectiveness of the provisions examined. They first of all identify that, although the term 'stakeholder' is referred to in the majority of the Codes, further analysis supports the point that the requirements in the Corporate Governance Codes are far from reflecting a uniform, comprehensive, strategic approach to stakeholders; this is because the term 'stakeholder' is understood and defined differently by each Code, and also because the Codes diverge on the significance they attach to the reference made [2]. Furthermore, as pointed out, disclosure related provisions regarding stakeholders can require either information *about* stakeholders or information *directed at* shareholders or both [2]. There is a distinct difference and purpose between the three options. Finally, another issue of importance relates to *what section* of the respective Code refers to the CSR disclosure requirements, as this also has an impact on whether CSR or sustainability is furthered. Reference to stakeholders is to be found in certain Codes, *not* in key principles and key sections, but in certain instances *only* in the Codes' preambles, forewords or prefaces [2]. The general problem with Corporate Governance Codes in the EU according to Szabó and Sørensen is that despite various existing Codes showing a great variety of solutions to the integration of CSR, CSR related recommendations are, in fact, not specific [2]. As a result, the identified vagueness makes CSR recommendations 'soft' and frequently open to interpretation [2].

Hence, in order to consider what type of provision would be most effective in supporting corporations' sustainable practices, focus needs to be given to the wording of the provisions, which needs to be specific. Consideration also needs to be given to which section the said provision would feature in, in the respective Corporate Governance Codes. The Danish Corporate Governance Code, the Dutch Corporate Governance Code, and the UK Corporate Governance Code with reference to 172 Companies Act 2006 are referred to further by the present article as useful paradigms to rely on to develop some alignment between the various Codes in the EU. The selection of the aforementioned Codes was conducted on the basis of using two stakeholder friendly Codes that have been identified to stand out among others for their support of sustainability, and using one, the UK Code, as one which is characteristic of the shareholder model, but which recently has been subjected to reform that supports an enlightened approach towards stakeholders.

One of the Codes that is singled out by the study of Szabó and Sørensen for its clear stakeholder policy recommendation towards company boards within its text is the Danish Code [2]. More specifically,

the Danish Code stands out, as it provides separate sections on the topic of "Communication and interaction by the company with its investors and other stakeholders," including "a dialogue between company, shareholders and other stakeholders" in Recommendation 1 [77]. It also provides a separate section, Recommendation 2 on "Tasks and responsibilities of the board of directors" [77] including a subsection numbered 2.2. on corporate social responsibility stating that: "The Committee recommends that the board of directors adopt policies on corporate social responsibility ... In this connection, the board of directors may take a position on the company's possible adoption of recognised national and international voluntary initiatives" [77]. It can be easily identified that the Code reinforces the need for communication with stakeholders and prompts boards to take on, as one of their responsibilities, to address a CSR policy with reference to national and international voluntary standards. This platform better aligns itself with the aforementioned initiatives, such as SDGs, ESG and Non-Financial Reporting initiatives.

The Dutch Corporate Governance Code also stands out for its stakeholder friendly approach. The new elements of the Dutch Code, following its reform in 2016, showcase that it now places greater emphasis on long-term value creation and risk management, and it introduces the notion of culture as a new element [78]. A key provision in the Dutch Corporate Governance Code [79], namely Principle 1.1., refers to the long- term value creation strategy of the company and imposes a responsibility of the management board to develop this [79]. The Code does not limit itself to the mention of this generic duty but gives guidance on how this can be effected in Definition 2.5., by stating what factors the board should pay attention to; two of which, among others, are namely: (a) stakeholders' interests, and (b) any other aspects relevant to the company and its affiliated enterprise, such as the environment, social and employee-related matters, the chain within which the enterprise operates, respect for human rights, and fighting corruption and bribery [79]. The notion of culture also features as a distinct provision in Principle 2.5.1 and is defined by the Dutch Code as: "the values that implicitly and explicitly inform employees' actions and the resulting behaviour"; a note is made, however, that the Code is not prescriptive as to exactly what culture is or should be, but that this is up to the management board to create it [79]. Principle 25 follows on to impose on the management board the responsibility of creating a culture aimed at long-term value creation for the company and its affiliated enterprise, with the supervisory board also being responsible of supervising the activities of the management board in this regard [79]. One of the strong aspects of the Dutch Code relates to the fact that it adopts a softer approach towards corporate governance and sustainability by focusing on the importance of culture, outside the strict realm of strategy. Another strong aspect of the Code is that it clearly outlines important key elements that allow for a better understanding of sustainability in the Environmental, Social and Governance context, as it refers to key words, such as: the environment, social and employee-related matters, the chain within which the enterprise operates, human rights, and corruption and bribery.

The UK constitutes an interesting paradigm to refer to considering that it is one of the Codes which is characterised by its shareholder-oriented approach, but which has recently, in 2017, been subject to reform in an attempt to better align itself with international and EU policy trends relating to sustainability. The Corporate Governance Code in the UK now, following it being subject to reform, refers to Section 172 Companies Act 2006 [80]. The reform provides a certain level of optimism that outsider Anglo-Saxon models of corporate governance come to similarly acknowledge the need for Corporate Governance Codes to, henceforth, better align themselves with sustainability by prompting the board of directors to pay respect to the interests of other stakeholders and sustainable practices during the board's decision-making processes. It also showcases that stakeholders' interests are an integral part of good governance, and not distinct from it. Principle 1 of the UK Code on Board Leadership and Company Purpose, among others, places focus on the board's responsibility to: " ... understand the views of the company's other key stakeholders and describe in the annual report how their interests and the matters set out in Section 172 of the Companies Act 2006 have been considered in board discussions and decision-making. The board should keep engagement

mechanisms under review so that they remain effective" [80]. Section 172(1) of the Companies Act 2006 is a hard law provision, which, however, has more of a soft law approach, considering that it is a duty which requires a high threshold to be reached in order to be enforced [81]. The duty is useful, however, as it prescribes the enlightened shareholder value approach to board decision-making, and outlines what the board of directors has a duty to promote for the success of the company for the benefit of its members as a whole, and outlines the factors that the board can have regard to (amongst other matters). Reference is made to the likely consequences of any decision in the long-term, the interests of the company's employees, the need to foster the company's business relationships with suppliers, customers and others, the impact of the company's operations on the community and the environment, the desirability of the company maintaining a reputation for high standards of business conduct, and the need to act fairly as between members of the company. Principle C.2. of the UK Code makes provision for Risk Management and Internal Control and prompts the board to identify, assess and monitor the principal risks that the company may be subject to in their annual report. Stakeholders' interests could be seen as part of, as well as separate to, this type of risk assessment, but the UK Code does not explicitly refer to stakeholders nor to other aims and objectives in this section. Additionally, Principle 1 of the Code, which refers to Section 172 CA 2006 as part of the annual report, merely confirms a pre-existing and acknowledged reporting requirement imposed by the Non-Financial Reporting Directive requirements. Section 414A CA 2006 has already required that the board of directors of companies to which the section applied to, produce a *Strategic Report* and outline its purpose, which was to inform members of the company and help them assess how the directors have performed their duty under Section 172.

The brief overview of three paradigms of Corporate Governance Codes, which serve as useful examples of provisions that make it an aim to support sustainability, each in a different way, showcase how the two former paradigms are more effective in doing so, and the latter paradigm not as much. The following section will aim to showcase the importance of investing in a soft law regulatory tool instead of a hard law one with the objective of nudging policy away from the shareholder primacy norm, which has dominated the scene since the 1970s.

*4.3. The Use of the Codes and Signaling: Theoretical Underpinning and Reasoning*

The present section aims to provide a theoretical underpinning relating to the reason why adopting particular provisions focused on sustainability in Corporate Governance Codes across EU Member States' Codes may be helpful in furthering sustainability goals. Both CSR and Corporate Governance have similarities in terms of their mode of regulation. The mode of regulation for both is that of a *soft law* approach [2]. Legal scholars have already identified that most EU Codes fail to truly support CSR, even though superficial mention of it in a wide range of paradigms can be found [2,3]. With reference to natural persons, Schneider and Ingram have helpfully identified a series of reasons that may exist if there is evidence that people are not taking actions needed to ameliorate social, economic, or political problems. These include that: (a) they believe that the law does not direct them or authorise them to take action, (b) they lack incentives or capacity to take the actions needed, (c) they may disagree with the values implicit in the means of the ends, or (d) the situation may involve such high levels of uncertainty that the nature of the problem is not known and it is unclear what people should do or how they might be motivated [82]. Furthermore, they explain that policy tools, including authority, incentives, capacity and symbolic and hortatory tools, can help address such issues [82]. Incentive tools, on the one hand, assume that individuals are utility maximisers and rely on tangible payoffs, positive or negative, to induce compliance or encourage utilisation [82]. Symbolic or hortatory tools, on the other hand, assume that people are motivated from within and decide whether or not to take policy-related actions on the basis of their beliefs and values [82]. Corporate Governance Codes could arguably be classified as one of the latter kinds of tools.

Hart provides a classification of different functions of the laws, namely that that the law can function either as a social construct, meaning a reflection of shared practices, or as a form of coercion/punishment

or as a signalling of appropriate behaviour, i.e., when the law has an expressive function [83]. Laws that come in the form of punishment are not to be preferred in the first instance according to Ayres and Braithwaite [84]. They argue that cooperative laws are better than strictly punitive ones, as research evidences that a punitive regulatory regime is likely to lead firms to circumvent the spirit of the law [84]. Hence, they propose that, as a first priority, regulatory problems should be solved by appealing to the social responsibility of firms, and if that fails, *then* revert to a responsive strategy through deterrence regulation [84]. Sunstein similarly refers to what is labelled as the *expressive function of law*, i.e., the function of law in "making statements" which is contrasted to the law which has the objective of controlling behavior directly by focusing on how legal statements might be designed to change social norms [85]. Sunstein's main argument similarly is that the *expressive function of law* should serve the purpose of managing social norms, and the law, in this way, should be effective in providing correctives [85]. Hence, the fact that Corporate Governance Codes of most EU Member States provide high levels of uncertainty relating to the nature of the problem, and uncertainty with regard to the exact course of action that corporations should take in relation to sustainability, reinforces the argument that reform is necessary. A first step is to proceed with reform of this soft means of regulation that will appeal to the social responsibility of firms, and it is only if that approach fails that one should proceed to explore more interventionist means of regulating. As Knapp explains, it is also the case of the soft law and its voluntary nature in the form of 'comply or explain' means that the Code can be reviewed when companies or shareholders feel it is appropriate [86]. Additionally, most countries will annually review companies' compliance with specific provisions of the Code. In the UK, for example, the latest annual review with the Financial Reporting Council's report has shown that the majority of companies declared themselves fully Code compliant and provided for strict compliance with the provisions of the Code, with associated problems only identified in relation to the lack of information on the outcomes of governance policies and practices [87]. This suggests that compliance in relation to the provisions of the Code is to be expected in the majority of the cases, which reaffirms that compliance is both followed and monitored by the regulatory authority.

### 4.4. Proposal for Reform

Policymakers are in the process of taking concrete steps towards enabling sustainability to form an integral part of good corporate governance. Metrics on sustainability, however, which would support the function of the market towards encompassing sustainability as a part of companies' performance, are currently unavailable in the form which would be reliable and could hold companies to account. Markets cannot yet 'value' sustainability [87]. The quality of the content of the monitoring of compliance and assessing the utility of information that form metrics are distinct issues outside the scope of the discussion of the present article. However, the quality of the provision supporting sustainability is, which leads us to the present section on proposals for reform.

The proposal for reform follows on from having considered the following key points when reviewing the sample of three Corporate Governance Codes above: (a) how are stakeholders defined in the Code? (b) what type of information is required with regards to transparency? (b) who does the information target (the markets, the board, the stakeholders or the shareholders)? (c) how powerful is the use of the words 'should', 'could' or 'must' when advising on any particular course of action and which is from a political or practical point of view preferable? (c) which section of the Code should this feature in? and (d) to what extent should the Code advise on transparency regarding information on the company's sustainability and impact on stakeholders, and on the engagement of stakeholders in the company's corporate strategy or decision-making process?

Firstly, Corporate Governance Codes need to provide a clear definition of the term 'stakeholder', which should include employees, clients, creditors, the communities in which they operate in or affect, suppliers, the environment, and vulnerable groups of people. Secondly, risk management should feature as a separate section in the Code and should not be used as a substitute for furtherance of sustainability through the protection of stakeholders. A separate section that focuses exclusively

on sustainability, with the potential of utilising the ESG and SDG paradigms should be adopted. The preamble, forward and preface of the Codes should not be used as a space to include the main provision on issues relating to sustainability and stakeholder protection. Thirdly, it is important to carefully consider the narrative of the text to be included in the proposed provision and, in this case, the Dutch Code, Danish Code and UK Code qualify as useful paradigms. The Dutch Code, specifically in its latest edition, provides a good narrative, insofar as it refers to the concept of long-term value creation and company culture. The terms 'stakeholders' and 'planetary boundaries' and 'risk management' are terms that should also be found and be defined at the outset of the respective Corporate Governance Codes in the section on definitions. The term 'planetary boundaries' should be understood within the planetary boundaries framework as defined by scientists, which identifies nine boundaries related to critical Earth system processes, and which jointly define a safe operating space [26,27].

The proposal put forward by the present article, building on the key points of the Danish, Dutch and UK Codes which have been referred to, is the following. A new provision, adopting all the above aforementioned characteristics should be included in a separate section which should be entitled and worded as follows:

Relations with Stakeholders as part of the board's responsibility

1.1. The board should adopt values for the company and its affiliated enterprise that contribute to a culture focused on long-term value creation and the board is responsible for the incorporation and maintenance of the values within the company and its affiliated enterprise. Attention must be paid, among others, to the following: i. the strategy and the business model; ii. the environment in which the enterprise operates; and iii. the existing culture within the enterprise, and whether it is desirable to implement any changes to this.

1.2. The board will ensure that a satisfactory dialogue with stakeholders takes place and that during the board's decision-making process, the board considers stakeholders' interests and develops, as well as complies with, a strategy that supports sustainable practices in the environmental, social and economic sense with respect to the planetary boundaries as defined by the scientific community. Specific consideration shall be given to:

- The impact of the company's operations on its stakeholders, the community and the natural environment;
- Aspects relevant to the company and its affiliated enterprise, such as the environment, social and employee-related matters, the chain within which the enterprise operates, respect for human rights, and fighting corruption and bribery.

## 5. Conclusions

The financial crisis that emerged in 2008, followed by the declared emergency climate crisis facing the planet in 2019, made clear that the regulation of sustainability within the corporate governance context needed to become a priority for policymakers and private market players alike. The present article supports the adoption of a policy that encourages a uniform regulatory approach in relation to sustainability, insofar as corporate governance needs to make clear that sustainability as a notion is internalized within what is considered as 'good practice. It is crucial to finally eliminate any trace of shareholder primacy policy still embedded in provisions that in theory aspire to support efforts towards sustainability. The shareholder primacy norm which had been driving the Commission's policy-making so far needs to be abandoned if any effective change is to be made in truly supporting corporations' sustainable practices. This is not limited to, but unquestionably includes, the Corporate Governance Codes of the respective EU Member States. A thoughtful and legitimate explanation for the adoption of a stakeholder friendly provision in the Corporate Governance Codes of all EU Member States is suggested as a nudge towards a change of policy outlook. The proposal is one which would better support recent initiatives at an EU and at an international level that aim to provide real support to sustainability concerns. Future research can follow on to explore means of

ensuring alternative forms of compliance with the provision of the Code, that go beyond the expected monitoring by the shareholder body. The addition of a provision in the EU Corporate Governance Codes that regulates the relationship of the company with stakeholders in a harmonised way may be an option that allows for better monitoring of companies' sustainable practices in the future, insofar as it could also help develop metrics for sustainable practices. The monitoring function could possibly be exercised by dedicated regulatory bodies or agencies with expertise on issues relating to sustainability. Such a framework could then align the information derived from the auditing of corporations on their sustainable practices with the information derived from their respective reporting on sustainability via non-financial reporting initiatives leading potentially to reliable sustainability metrics.

**Funding:** This research received no external funding.

**Acknowledgments:** The author wishes to thank all the members of the Sustainable Market Actors for Responsible Trade Project Horizon 2020, and especially the primary investigator of the project, Professor Beate Sjåfjell, for supporting efforts to research in the field of sustainability and company law reform.

**Conflicts of Interest:** The author declares no conflict of interest.

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
