# Peer review of "A Proposal for Reform of EU Member States’ Corporate Governance Codes in Support of Sustainability"

_sustainability, doi:10.3390/su12104328_

Round 1
Reviewer 1 Report
The article makes an important argument about the need for coherence between various forms of business regulation for sustainability (e.g. reporting and corporate governance), suggests a reform (a common EU code of Corporate Governance) and suggests some elements of such a code.
However, the logic of the argument (how one point relates to the next; how one bit of evidence or analysis supports a point being made) suffers from problems of style that make it very difficult for the reader to understand. The main problems are 1. Long paragraphs that seek to do too much and therefore confuse the reader about the point being made; 2. Run-on sentences, which can be found in almost every paragraph or at least every page, making it very difficult to know what is the main point being made; 3. Unclear headings, in particular the use of catch phrases (e.g. "best practice", "grass is greener") that do not provide guidance to the reader about what that section is about. All of these can be fixed by revising the paper as follows: 1. Stick to simple paragraph structure - Ensure each para has an introductory sentence with one main point, and that the other sentences elaborate or evidence that point; 2. Sentences should make only one point; 3. Break up sections into sub-sections, with simple headings, and ensure that these tell the story / make the argument you want to make; 4. consider having in each section a brief one or two sentence conclusion, one that reflects the heading of that section; 5. be consistent in language used, i.e. avoid introducing new concepts in concluding sentences of paragraphs or sections (e.g. the role of risk management is only mentioned in the 6. proposal for reform, and it is not clear what part of the reform proposal it relates to), or new terms or phrases to describe things already discussed (e.g. accountability appears in the conclusion but it is not clear how it relates to the reform proposal)
The result of these problems of style is a difficulty for the reader in understanding the relevance of certain sections: e.g. section 3.2.1 "Other initiatives" is interesting but the reader is left unsure how it relates to the overall point; the same is true of 3.2;
Similarly, but more important, in the conclusion it is not clear to the reader why exactly a "stakeholder friendly provision" in CG codes would improve oversight, and by whom. This is a central point in the argument but the problems of style means it is lost to the reader. In short, in what way would a SH friendly provision improve oversight and where in the article is this point made?
Perhaps consider moving the reform proposals higher up and structuring the rest of the paper along the lines of the reform proposals, with text under each point of the proposal explaining what the proplem is that reform point seeks to address and how it solves or addresses it.
Author Response
Stick to simple paragraph structure - Ensure each para has an introductory sentence with one main point, and that the other sentences elaborate or evidence that point; THIS HAS BEEN ADDRESSED Sentences should make only one point; THIS HAS BEEN ADDRESSED Break up sections into sub-sections, with simple headings, and ensure that these tell the story / make the argument you want to make; THIS HAS BEEN ADDRESSED Consider having in each section a brief one or two sentence conclusion, one that reflects the heading of that section; THIS HAS BEEN ADDRESSED Be consistent in language used, i.e. avoid introducing new concepts in concluding sentences of paragraphs or sections (e.g. the role of risk management is only mentioned in the proposal for reform, and it is not clear what part of the reform proposal it relates to), or new terms or phrases to describe things already discussed (e.g. accountability appears in the conclusion but it is not clear how it relates to the reform proposal) THIS HAS BEEN ADDRESSED The result of these problems of style is a difficulty for the reader in understanding the relevance of certain sections: e.g. section 3.2.1 "Other initiatives" is interesting but the reader is left unsure how it relates to the overall point; the same is true of 3.2; THIS HAS BEEN ADDRESSED
7.Similarly, but more important, in the conclusion it is not clear to the reader why exactly a "stakeholder friendly provision" in CG codes would improve oversight, and by whom. This is a central point in the argument but the problems of style means it is lost to the reader. In short, in what way would a SH friendly provision improve oversight and where in the article is this point made? Have used signalling and nudge theory to support this point, as well as the tensions show how the Code is a soft law tool that could better accommodate EU tensions in this area of regulation.
Perhaps consider moving the reform proposals higher up and structuring the rest of the paper along the lines of the reform proposals, with text under each point of the proposal explaining what the problem is that reform point seeks to address and how it solves or addresses it. I HAVE DONE THIS AS SUGGESTED. THANK YOU FOR THE SUGGESTION. VERY HELPFUL.Reviewer 2 Report
The paper is founded on the premise of an incoherence in the EU's corporate governance (CG) framework. While it could be argued that the EU's CG framework remains a matter of academic interest despite the many studies which have already been conducted on the subject, the paper's proposition of an incoherence in the CG structure is not borne out by any evidence presented by the authors. Rather, the author's proposition of an incoherence is founded on an assumption, supposition or conjecture which is unsupported by any objective factual evidence or evidence drawn from secondary literature sources. this fact alone represents a major flaw with the paper.
Secondly, the paper set out as its main task to examine the question of sustainability within the framework of CG in relation to financial markets (see line 83 where the paper's contribution is stated as "... prism of coupling sustainability with financial markets law". The rest of the paper bears no relationship to this stated aspiration. This represents a second major flaw.
Much of the paper is descriptive (rather than analytical), and the discussion often tends to deviate from the stated aims and objectives. There is an uncritical use of concepts involving principles borrowed from other disciplines such as behavioural economics which normally applies to individuals (natural persons) and which the authors imports into corporate governance without (at the very least) questioning, explaining or even justifying the relevance of its application to corporate or legal persons.
The recommendation or proposals posited by the authors (none of which is directly linked to the issue of sustainability in financial markets law) offer nothing new and is simply a re-hatching of generic ideas and concepts.
There are are some obvious errors of judgement: for example, The Climate Change agreement is presented as having "proven to be a great success", whereas the evidence clearly points to the contrary as seen recently with the Madrid summit which ended in failure as well as the increasing militancy of extinction activists.
On the whole the paper is poorly conceived and is based on a foundation of assumption and supposition with no reference to objective factual or scholarly evidence. It is poorly presented in many places (see attached text for more feedback comments), fails to meet its own key aspiration of "coupling sustainability with financial markets law", and the arguments are very weak on the whole.
The flawed foundations on which the paper is based (an assumed rather than proven incoherence in the EU's CG framework), together with the non-realization of the proposed contribution (coupling sustainability with financial markets law) raises serious questions as to whether this a really a viable research topic.

Author Response
1.The paper's proposition of an incoherence in the CG structure is not borne out by any evidence presented by the authors. Rather, the author's proposition of an incoherence is founded on an assumption, supposition or conjecture which is unsupported by any objective factual evidence or evidence drawn from secondary literature sources. this fact alone represents a major flaw with the paper. This is a good point. The paper now has abandoned what had been labelled as incoherence, but rather shows through section 2 now that the EU has been advocating for supporting stakeholders and sustainability, without taking on any serious steps towards that direction, albeit in a few initiatives. The tensions of why this is the case have been outlines, and also a way forward has been proposed in section 2.
2.Secondly, the paper set out as its main task to examine the question of sustainability within the framework of CG in relation to financial markets (see line 83 where the paper's contribution is stated as "... prism of coupling sustainability with financial markets law". The rest of the paper bears no relationship to this stated aspiration. This represents a second major flaw. This has also been abandoned as the point here is right. The research question set out now hopefully clarifies what the central aim of the article is.
3.Much of the paper is descriptive (rather than analytical), and the discussion often tends to deviate from the stated aims and objectives. There is an uncritical use of concepts involving principles borrowed from other disciplines such as behavioural economics which normally applies to individuals (natural persons) and which the authors imports into corporate governance without (at the very least) questioning, explaining or even justifying the relevance of its application to corporate or legal persons. These points have been resolved by restructuring the paper and following steps to answer the main research question set out in the beginning. Behavioural economics point has also been removed.
4.The recommendation or proposals posited by the authors (none of which is directly linked to the issue of sustainability in financial markets law) offer nothing new and is simply a re-hatching of generic ideas and concepts. Has been omitted
5.There are some obvious errors of judgement: for example, The Climate Change agreement is presented as having "proven to be a great success", whereas the evidence clearly points to the contrary as seen recently with the Madrid summit which ended in failure as well as the increasing militancy of extinction activists. Has been omitted
6.On the whole the paper is poorly conceived and is based on a foundation of assumption and supposition with no reference to objective factual or scholarly evidence. It is poorly presented in many places (see attached text for more feedback comments), fails to meet its own key aspiration of "coupling sustainability with financial markets law", and the arguments are very weak on the whole. Has been largely restructured and have added literature and revised approach.
7.The flawed foundations on which the paper is based (an assumed rather than proven incoherence in the EU's CG framework), together with the non-realization of the proposed contribution (coupling sustainability with financial markets law) raises serious questions as to whether this a really a viable research topic. Has been largely restructured and have added literature and revised approach.
Reviewer 3 Report
Title needs to be more focused and shorter.
There is no visible research problem, just interest to map and decode the field. Author should from the beginning offer more attention to problem solving nature of study. Improving existing bureaucracy and making it sustainable might seem important, but more visible connection with actual socio economic problems would be appreciated.
Focus on sustainability builds good connection with journal focus.
Abstract is relatively long and not succeeding to point the focus of study, please shorten and make more focused.
In introduction a lot of attention is given to sustainability and efficiency aspects it is hard to understand why sustainability and efficiency are useful for societies and economies (not everything needs to be preserved and survived).
More effort is needed to make central hypothesis understandable for reader.
Relevant reference to key concepts and termins are missing
Introduction also gets to details while there is still much higher demand for explanations about importance and added value of current study.
It would be also appreciated to add paragraph about research methodology into introduction. In current form reader expects the study to remain rather descriptive.
There are no clear definitions and statements of terminology.
2nd chapter is long sophisticated but very descriptive without any research focus or research question to be answered. Reviewer would suggest to shorten it by 20% and add a section why this overview is needed and how it contributes to the research focus and solving some relevant problems. Sustainability itself is not always a virtue, for example there would be rather little value in sustainability of Soviet Union and ruling of Brezhnev in Soviet Union.
Author uses mostly non-measurable termins which sound positive but provide no scientific bases for critical assessment.
Third chapter starts to bring in factors, which however have not been methodologically integrated and justified, they might have relevance for study but author needs to bring it to reader.
Author needs more integrate the empirical information to study focus.
Conclusions spend a lot of words for declarative statements, which however are not summarizing the article and in many cases have distant relations with research focus.
In generally manuscript is more philosophical essay or compendium than methodological academic study. Author cherry-picks suitable examples and quotations, but there is lack of systematic approach and critical aspects.
What is the scientific added value of the study?
Descriptive parts of study and language are in good level, more referencing is needed with key terminology and concepts.
This manuscript can hardly be published without any reasonable methodological component with for empirical selection and analysis.
Author Response
Title needs to be more focused and shorter. Has been altered. There is no visible research problem, just interest to map and decode the field. Author should from the beginning offer more attention to problem solving nature of study. Improving existing bureaucracy and making it sustainable might seem important, but more visible connection with actual socio-economic problems would be appreciated. Good point. A research question has been introduced. Focus on sustainability builds good connection with journal focus. Abstract is relatively long and not succeeding to point the focus of study, please shorten and make more focused. I have thank you. In introduction a lot of attention is given to sustainability and efficiency aspects it is hard to understand why sustainability and efficiency are useful for societies and economies (not everything needs to be preserved and survived). More effort is needed to make central hypothesis understandable for reader. Relevant reference to key concepts and terms are missing. See section on definitions and tensions (Section 3) Introduction also gets to details while there is still much higher demand for explanations about importance and added value of current study. It would be also appreciated to add paragraph about research methodology into introduction. In current form reader expects the study to remain rather descriptive. There are no clear definitions and statements of terminology. 2nd chapter is long sophisticated but very descriptive without any research focus or research question to be answered. Reviewer would suggest to shorten it by 20% and add a section why this overview is needed and how it contributes to the research focus and solving some relevant problems. Sustainability itself is not always a virtue, for example there would be rather little value in sustainability of Soviet Union and ruling of Brezhnev in Soviet Union. This has been amended and tensions section, as well as narrative has been added. Author uses mostly non-measurable termins which sound positive but provide no scientific bases for critical assessment. Third chapter starts to bring in factors, which however have not been methodologically integrated and justified, they might have relevance for study but author needs to bring it to reader. Have revised structure to explain relevance. Author needs more integrate the empirical information to study focus. Existing literature mentioned in the introduction has been expanded on in proposal for reform section. Conclusions spend a lot of words for declarative statements, which however are not summarizing the article and in many cases have distant relations with research focus. Focus altered. Transitioning paragraphs explain thread connecting all now. In generally manuscript is more philosophical essay or compendium than methodological academic study. Author cherry-picks suitable examples and quotations, but there is lack of systematic approach and critical aspects. Made evident in abstract intro and conclusion. What is the scientific added value of the study? Made evident in abstract intro and conclusion. Descriptive parts of study and language are in good level, more referencing is needed with key terminology and concepts. Have been added. This manuscript can hardly be published without any reasonable methodological component with for empirical selection and analysis. Part 2 has attempted to address some of these issues. Additional comments on how to improve at stage 2 very welcome.
Round 2
Reviewer 2 Report
The author has addressed the key concerns from the previous review through extensive restructuring of the paper and modification of the paper's title in order to reflect these changes.
I would recommend further proofreading before publication, with particular attention paid to the following:
too many keywords: I still think the there is very little in the paper on financial markets law, and I would suggest that this particular keyword be deleted as it could be misleading to readers. lines 81-82 need checking/proofreading line 105: suggest removal of the word originality; originality in research can come from either primary data collection and analysis, or through conceptual development of the subject matter from identification of the research problem to the formulation of a solution (e.g. in the form of a conceptual framework) - not from simply recycling data from previous literature sources, which is what the author seems to be suggesting. This line should now read: "The present paper ...." lines 136-138: does not constitute a proper sentence (needs checking). line 888: does the author mean 'theoretical underpinning' (rather than "theoretical pinning")? line 971: the author refers to "planetary boundaries": what does this actually mean? line 978: need checking/ proofreading. line 987: in which the author refers to the monitoring of CG Codes by shareholder bodies; but shouldn't the monitoring function be exercised by dedicated regulatory bodies or agencies rather than by shareholders? line 989: needs checking/ proofreading.Author Response
I am very grateful to the reviewer for making suggestions for further improvement. Attempts have been made within this short time period to address all of the issues below.
Reviewer 3 Report
Abstract in current form is very long and would benefit from shortening and additional focus. Also some acronyms appear in abstract without original form.
Starting from introduction (but in other parts of article as well), some paragraphs need shortening or division into two parts.
Introduction consists also some descriptive information, which would fit better to following chapters of study.
Main body: the historical overviews are occupying large part of study while having limited added value to research aim.
Conclusion, as being almost 3 times shorten than abstract, clearly needs additional strong content and message.
References: most of citations cover the full range of sources, but it would be preferred to refer to specific pages, which cover the referred content.
Reform proposals are the second part of an article which would benefit from additional strong content
To conclude:
- Article would strongly benefit from shortening in terms of quantity but also removing some less relevant information not contributing to answering research question but distracting the reader´s attention.
- Article in current form remains descriptive and debative without having strong methodological backbone, however in current form it is hardly possible to add it. What the author of course can do is to integrate different parts of study more visibly to research aim, by adding clarifying sentences to the beginning of sub-chapters.
Author Response
Thank you to reviewer 2 who has addressed the issues that require attention in a thematic order. I have reorganized several sections and made changes to the abstract, main body, proposals for reform as best as the limits time frame would allow and hope it may serve as a useful improvement of the overall points raised in the article.